# Genetic and Evolutionary Analysis of Ake Chicken: New Insights into China’s Sole Indigenous Naked-Neck Chicken Breed

**DOI:** 10.3390/ijms26094399

**Published:** 2025-05-06

**Authors:** Ronglang Cai, Shuang Gu, Boxuan Zhang, Xuemei Deng, Mostafa Galal Abdelfattah, Ning Yang, Hesham Y. A. Darwish, Congjiao Sun

**Affiliations:** 1Frontier Science Center for Molecular Design Breeding, State Key Laboratory of Animal Biotech Breeding, China Agricultural University, Beijing 100193, China; cairl@cau.edu.cn (R.C.); 13592566171@163.com (S.G.); zhangbx@cau.edu.cn (B.Z.); deng@cau.edu.cn (X.D.); nyang@cau.edu.cn (N.Y.); 2National Engineering Laboratory for Animal Breeding and Key Laboratory of Animal Genetics, Breeding and Reproduction, Ministry of Agriculture and Rural Affairs, Department of Animal Genetics and Breeding, College of Animal Science and Technology China Agricultural University, Beijing 100193, China; 3Department of Poultry Production, Faculty of Agriculture, Assiut University, Assiut 71526, Egypt; mostafagalal@aun.edu.eg; 4Department of Applied Biotechnology, Molecular Biology Researches & Studies Institute, Assiut University, Assiut 71526, Egypt

**Keywords:** Ake chicken, population phylogenetic and structure analysis, gene flow, effective population size, population separation history

## Abstract

Heat-stress resilience is vital for poultry in tropical/subtropical regions where high temperatures impair productivity. Ake chickens, as the only naked-neck chicken breed in China, exhibit robust resistance to heat stress, but this breed lacks clarity in its genetic origins. This study utilized the next-generation sequencing data from 22 chicken breeds to conduct phylogenetic and population analyses. Gene flow analysis revealed a gene migration event from Iranian naked-neck chickens and Indian local breeds to Ake chickens, and population separation estimates suggested that the naked-neck gene was introduced to China around 500–600 years ago. NJ-tree, PCA, and population structure analyses showed that Ake chickens cluster with Yunnan native breeds, which diverged only 100–200 years ago. A selective sweep in the candidate region on chromosome 3 (97.0–97.37 Mb) showed elevated genetic differentiation (F_ST_) and reduced nucleotide diversity (π) compared to the genome-wide average, indicating rapid fixation of the trait under natural/artificial selection. Demographic reconstruction indicated that the current effective size of Ake chickens is stable at 2000–3000 individuals. These findings deepen our understanding of Ake chicken evolution and provide valuable insights for conservation and the development of heat-stress-resistant poultry breeds.

## 1. Introduction

Poultry, being a cost-effective and high-quality protein source, plays a crucial and irreplaceable role in animal husbandry. Due to the absence of sweat glands, chickens are particularly sensitive to the rearing environments characterized by their high temperature and humidity [1], which can significantly impact their production performance. In recent decades, modern poultry breeding has led to the development of fast-growing breeds with higher metabolic rates, which in turn results in increased thermogenesis. In tropical and subtropical regions, high temperatures have severely affected the poultry industry [2]. In 1992, Cahaner conducted feeding experiments with broilers under both high- (33 °C) and normal- (20 °C) temperature conditions [3], and demonstrated that chickens are unable to efficiently metabolize heat, resulting in a reduced feed intake, decreased weight gain, and diminished egg production. Therefore, it is important to enhance the heat-dissipation capacity of poultry.

The application of the naked-neck trait primarily demonstrated resistance to heat stress. Compared to wild-type individuals, the feather production of homozygous and heterozygous naked-neck chickens was reduced by approximately 40% and 25%, respectively. This reduction facilitates enhanced heat dissipation, thereby improving production performance, including egg number, egg quality, and meat yield, in environments characterized by their high temperature and humidity [4]. Furthermore, other studies have indicated that homozygous naked-neck chickens display notable enhancements in reproductive characteristics, such as semen volume and sperm count, across all ages [5]. In high-temperature environments, the proportion of abnormal embryos of naked-neck chickens is markedly reduced compared to normal-feathered chickens, suggesting that the embryos of naked-neck chickens exhibit greater vitality and survival rates [5].

China has rich and diverse chicken breeds. The Ake chicken, native to Fugong County in Yunnan Province, China, is a dual-purpose breed that is valued for both meat and eggs. A distinctive feature of the Ake chicken is its naked neck (Figure 1), which also results in fewer body feathers. Notably, the Ake chicken is the only breed in China that exhibits the naked-neck trait. It has been reported that the naked-neck mutation in Ake chickens is characterized by a 73 kb insertion at the end of chromosome 3 (chr 3), which is derived from chromosome 1 (chr 1), and it is a chromosomal duplication [6]. In addition, Ake chickens display considerable variations in their appearance (like diverse feather colors and comb types), resembling many other Chinese native breeds. This indicates that Ake chickens may have undergone complex gene flow and population hybridization, so it is necessary to investigate the population development history of Ake chickens.

In this study, we performed whole-genome sequencing of Ake chickens and Egyptian naked-neck chickens, and integrated the data with next-generation sequencing (NGS) data from chicken breeds across China, India, Egypt, and Iran to investigate the genetic background and population divergence history of the Ake chicken. This study provides a valuable scientific foundation for the breeding of heat-stress-resistant chicken breeds.

## 2. Results

### 2.1. Phylogenetic Relationships

The phylogenetic relationships and population genetic structure were analyzed using a total of 297 individuals belonging to 22 breeds from various regions (Table 1). A phylogenetic NJ tree was constructed, which illustrated that the Ake chicken shared a branch with several Yunnan local breeds, including the Nixi chicken (NX), Yimen chicken (YM), Zhaotong chicken (ZT), Zhenyuan chicken (ZY), and Wuliangshan chicken (WLS), followed by breeds from southern China, Indian local breeds, the RJF, the Egyptian naked-neck chicken (EgyptNa), and the Iranian naked-neck chicken (IranNa) (Figure 2A). The PCA result was further utilized to evaluate the population structure, with the first three principal components explaining 19.18%, 15.86%, and 14.31% of the variance, respectively (Figure 2B). Each group displayed distinct clustering patterns. The findings suggested a close relationship between Ake chicken and local breeds in Yunnan province, while the Tibetan chicken, Indian breeds (India Local 1, India Local 2) and the IranNa were relatively closer to this category. In contrast, the Chahua chicken (CH), Huiyang beard chicken (HYB), and EgyptNa appeared to be more distantly related.

### 2.2. Population Structure

To determine the potential genetic admixture among these chicken breeds, we conducted a population structure analysis using the admixture method, a full maximum likelihood approach that estimates individual ancestry and admixture proportions, while assuming K (ranging from 2 to 25) ancestral populations (Figure 3A,B). When K was set to 5, we observed the emergence of differences between the Ake chickens and other breeds. As the K value increased, the genetic background differences among the various breeds became progressively more pronounced. Cross-validation error (Figure 3A) declines steeply from K = 2 to K = 7, reaching a global minimum at K = 7. For K > 7, errors plateau or increase slightly, indicating no further improvement in the model fit. Hence, K = 7 was selected as optimal, at which point the genetic composition of Ake chickens was found to be similar to that of Yunnan native chickens (Figure 3C). Additionally, Indian local chickens and the IranNa shared certain genetic components with the Ake chicken. In contrast, the EgyptNa exhibited a distinctly different genetic background from that of the Ake chicken, which aligns with the results of the principal component analysis.

### 2.3. Selective Sweep in the Naked-Neck Trait Locus

To further evaluate the genetic differentiation of Ake chickens from other breeds in the candidate region associated with the naked-neck trait, we focused on the chr 3 region (97.0–97.37 Mb) based on prior evidence from Cai et al. (2025), which identified a 73 kb duplication in this interval as the causal mutation for the naked-neck phenotype in Ake chickens [6]. We performed F_ST_ analyses comparing Ake chickens with other breeds across the entire genome and specifically within the 97.0–97.37 Mb interval on chr 3. As is shown in Figure 4, the F_ST_ values between Ake chickens and Egyptian naked-neck breeds in this candidate region were notably lower than the genome-wide average (0.04 vs. 0.09). On the contrary, the F_ST_ values between Ake chickens and Yunnan breeds in this region were notably higher than the genome-wide average, and the F_ST_ values in this region were 2–5 times higher than in the genome-wide range, suggesting stronger selection or genetic drift in this region. What is more, the nucleotide diversity (π) of Ake chickens in this region (0.1421) was significantly lower than the genome-wide average (0.2953), supporting the notion of a selective sweep.

### 2.4. Gene Flow Analysis

The analysis of phylogenetic relationships and the population genetic structure revealed extensive population mixing between Ake chickens and local chickens from Yunnan, the IranNa, and Indian breeds. To further elucidate admixture events and enhance our understanding of phylogenetic relationships, we employed TreeMix to construct a maximum likelihood (ML) tree using the RJF as the outgroup (Figure 5A). The resulting ML tree categorized these populations into three clusters, which aligned with the population structuring patterns identified in the population genetic structure analysis. Notably, up to 99% of the variations among species could be explained by a model incorporating four migration events. In this model, we observed a migration edge from the IranNa/India Local 2 to the Ake chicken. In addition, we found that migration events occurred between the IranNa and Egyptian naked-neck chicken, which was consistent with the results of admixture.

Further D-statistics revealed significant values (|Z| > 3,) with positive D values (Figure 5B). These results suggested a closer relationship between the Ake chicken and IranNa, followed by India 2 and BG, aligned with the findings of TreeMix.

### 2.5. Effective Population Size and Population Separation History Estimation

To further elucidate the breed origin of the Ake chickens, we conducted population history analysis. Similar methods such as PSMC and MSMC have been widely used to infer population size changes and segregation histories in humans and animals from multiple genomic sequences [7]. The SMC++ analysis indicated that the population size of the ancestors of the Ake chicken reached its peak approximately 100,000 years ago, followed by a continuous decline (Figure 6A). Unlike other breeds, which began to rebound in population size between 10,000 and 1000 years ago, the decline in the effective population size of Ake chickens continued until several decades ago and has only recently stabilized, with some signs of growth. The current effective population size of Ake chickens is about 2000–3000 individuals.

We subsequently conducted a population separation analysis and discovered that over the last 100 to 1000 years, Ake chickens have progressively diverged from other chicken breeds, with an earlier divergence from other countries’ chicken breeds (Figure 6B). Figure 6B illustrates that the separation of the Ake chicken from the IranNa and EgyptNa occurred approximately 500 to 600 years ago, while most separation events between Ake chickens and Yunnan breeds occurred within the past 100 to 200 years. This suggests that the naked-neck trait likely entered China a considerable time ago.

## 3. Discussion

With the advancements in genome sequencing technology, individual genomic information can now be directly used for kinship identification. SNPs have been extensively utilized in the investigation of population structure and genetic diversity across various common domestic animals, including pigs [8,9], chickens [10,11,12], cattle [13,14], and sheep [15,16].

China hosts a diverse range of local chicken breeds, each of which is recognized as a valuable genetic resource because of its unique traits and characteristics. Heterozygous occurrence among various chicken populations plays a crucial role in fostering genetic diversity. Furthermore, gene exchange and introgression contribute new genetic material to the original population, thereby facilitating alterations in specific traits. It is widely acknowledged that domestic chickens trace their origins to the red jungle fowl found in Southwest China [17], Southeast Asia, and other areas. Additionally, several researchers have noted that the genomes of numerous indigenous Chinese chickens have undergone introgression with genes from foreign commercial broiler chickens [18]. Given that Ake chickens exhibit a range of non-uniform phenotypes, including variations in feather colors, comb types, and skin colors, which are similar to those found in chicken breeds from Southwest China, we propose that Ake chickens are likely influenced by genetic introgression from Chinese native breeds. The results of population structure and TreeMix confirmed this.

Our genome-wide and region-specific F_ST_ analyses provide critical insights into the evolutionary dynamics of the naked-neck trait in Ake chickens. The notably lower F_ST_ values between Ake chickens and other naked-neck breeds within the candidate region (chr3: 97.0–97.37 Mb) compared to the genome-wide average (Figure 6) suggest conserved genetic architecture at this locus. Conversely, the elevated FST between Ake chickens and Yunnan local breeds in this region implies strong local adaptation. Meanwhile, the nucleotide diversity (Pi) in this candidate region is substantially reduced (0.1421) compared to the overall genome-wide value (0.2953). This pronounced reduction in genetic variation is consistent with a selective sweep, suggesting that once the naked-neck allele was introduced, it rapidly increased in frequency under both natural and artificial selection. These findings collectively suggest that the naked-neck trait in Ake chickens resulted from ancestral introgression of the allele followed by rapid fixation through local adaptation.

Furthermore, results from effective population size and population segregation history analyses suggest that the divergence of Ake chickens from Yunnan local breeds and Southeast Asian breeds occurred later than its divergence from other naked-neck breeds, coinciding with the peak activity of the Southwest Silk Road [19]. This finding aligned with the TreeMix result. The inferred gene flow from Iranian naked-neck chickens and Indian local breeds into Ake chickens (Figure 5A) likely reflects historical exchanges along the Southwest Silk Road, which connected Yunnan to South and West Asia. The naked-neck allele may have been introduced to Yunnan via trade routes linking Iran or India, but sustained gene flow was likely hindered by geographic barriers such as the mountains and rivers. Despite these introgressions, the Ake chicken genome remains predominantly influenced by Yunnan local genetic components, suggesting that only adaptive loci (such as naked-neck alleles) were retained after introgression occurred, while other foreign alleles were purged due to a mismatch with local environmental pressures. This pattern aligns with the “incomplete introgression” model observed in domestic chicken domestication [18]. Subsequently, local farmers likely accelerated the fixation of the naked-neck trait through phenotypic preference, like heat-resistance, ultimately leading to the formation of the Ake chicken. This is evidence exemplifying the synergy between artificial and natural selection: directional human selection fixed the trait, while natural selection preserved locally adaptive genes that are critical for surviving in a humid subtropical climate.

As the sole genetic resource possessing the naked-neck trait in China, the Ake chicken population is critically endangered, with only a few thousand individuals remaining. While their effective population size has stabilized in recent decades, population decline heightens inbreeding risks. To address this, we propose establishing a core breeding population by prioritizing individuals with high heterozygosity, while integrating the traditional free-range practices of local farmers to ensure the preservation of both genetic diversity and adaptive traits under natural environmental pressures.

## 4. Conclusions

In this study, we used WGS data to provide a new insight into the genetic origins and evolutionary history of the Ake chicken, the only indigenous naked-neck breed in China. Our findings revealed that the naked-neck trait was first introduced to China from Iran approximately 500 to 600 years ago. After nearly 200 years of multiple hybridizations with Chinese native breeds, especially the Yunnan local breeds, this trait was retained due to its favorable adaptability to the local climate and environment. This study will contribute valuable insights for future conservation strategies. Furthermore, it will provide a theoretical foundation and reference for the breeding of poultry breeds with enhanced resistance to heat stress, which is particularly significant in the context of global climate change.

## 5. Materials and Methods

### 5.1. Ethical Statement

Ake chickens were collected from Fugong County, Yunnan, China. Egyptian naked-neck chickens were sampled from the Poultry Research Farm of Assiut University, Egypt. All samples were obtained with explicit permission from the institutional authorities. No privately owned animals were involved in this study.

Ethical approval was obtained from the Animal Welfare Committee of China Agricultural University (AW71802202-1-2) and performed in accordance with the procedures of the Guide for the Care and Use of Laboratory Animals (China Agricultural University). Blood samples (2 mL) were collected from the wing veins of the experimental chickens, no animals were sacrificed and anaesthetized in this study. All methods are reported in accordance with the ARRIVE guidelines for the reporting of animal experiments.

### 5.2. Sample and Datasets

In this study, blood samples from 30 Ake chickens from Fugong County, Yunnan Province, and 30 naked-necked chickens from Egypt were used for genome sequencing. Additionally, we collected the NGS data of 237 chickens from 22 breeds available in public databases (Appendix A), including local breeds from Southwest China, Iranian naked-neck chickens, and Indian local breeds.

### 5.3. DNA Extraction

Two-milliliter blood samples from 30 Ake chickens and 30 Egyptian naked-neck chickens were obtained from the wing vein and then stored at −20 °C. DNA extraction was performed using the TIANamp Genomic DNA Kit (TIANGEN, Beijing, China) according to the manufacturer’s protocol.

### 5.4. Whole-Genome Sequencing, Reads Mapping, and Single-Nucleotide Polymorphism (SNP) Detection

At least 1 mg of genomic DNA per individual was used to construct a sequencing library according to the manufacturer’s specifications for the TruSeq Nano Sample Prep Kit (Illumina Inc., San Diego, CA, USA). Briefly, DNA was fragmented, end-polished, A-tailed, and ligated with a full-length adapter. Fragments of 400–500 bp were selected, amplified, and purified using the AMPure XP system (Beckman Coulter, IN, USA). The prepared libraries were assessed via an Agilent2100 Bioanalyzer and quantified via real-time PCR. Paired-end libraries with an average insert size of approximately 350 bp were sequenced on an Illumina HiSeq X Ten platform (Illumina) by Berry Genomics Co., Ltd., (Beijing, China), yielding 150 bp reads with a target depth of 10-fold coverage per genome. Sequence reads were filtered via fastp (v 0.12.3) [20], which discards reads with lengths of < 50 bp and N bases > 6. The high-quality trimmed read pairs were mapped on the reference genome GRCg7b (https://ftp.ensembl.org/pub/release-110/fasta/gallus_gallus/dna/, accessed on 1 May 2025) via BWA-MEM (v 0.7.17) [21], and the resulting alignment files were sorted using SAMtools [22]. The duplicated reads were marked via MarkDuplicates, and base quality scores that estimated their biases were recalibrated using the BaseRecalibrator and ApplyBQSR command in GATK (v 4.1) [23]. SNPs were detected via the HaplotypeCaller command, and the output gVCF files were merged via the CombineGVCFs command. The SNPs were filtered via the VariantFiltration and SelectVariants commands with the following criteria: QUAL < 40.0, QD < 2.0, FS > 60.0, MQ < 40.0, MQRankSum < −12.5, and ReadPosRankSum < −8.0. Finally, we retained biallelic SNPs in autosomes, ensuring a maximum missing data threshold of less than 0.90 and a minor allele frequency exceeding 0.05. A total of 23,749,617 SNPs were detected. After quality control and LD screening, 1,022,355 SNPs remained.

### 5.5. Population Phylogenetic and Structure Analysis

Plink (v 1.90) [24] was used to transform VCF files and for principal component analysis (PCA). The neighbor-joining (NJ) tree was constructed using VCF2Dis (v 1.46) (https://github.com/BGI-shenzhen/VCF2Dis, accessed on 1 May 2025), and visualized using FastME (v 2.0) [25] and online tool (https://itol.embl.de/, accessed on 1 May 2025). Population structure was evaluated using admixture (v 1.3) [26], with the analysis encompassing 2–8 genetic clusters (K = 2–25).

### 5.6. Detection of Selection Signatures

To identify genomic regions under selection for the naked-neck trait, we computed the nucleotide diversity (Pi) of Ake chickens and the population differentiation index (FST) between Ake chickens and other breeds using VCFtools (v 0.1.16) [27].

### 5.7. Migration Events and Admixture Analysis

To examine the migration and admixture of these populations, separation and mixture graphs were generated using TreeMix (v. 1.13) [28]. In these analyses, the red jungle fowl (RJF) was employed as an outgroup, with gene flow migration events ranging from 0 to 10 (m = 0–10), and each migration scenario was iterated 10 times. Additionally, D-statistics [29] were used to quantify introgression among populations, with calculations performed using the Dsuite program (v 0.5) [30]. RJF served as the reference population, with Ake chickens as the target population. We assumed the fixed ancestral allele in the reference population, and |Z| > 3 was used as a threshold to indicate significant gene flow from other groups into the target population. The result was visualized by an online tool (https://hiplot.com.cn/home/index.html, accessed on 1 May 2025).

### 5.8. Effective Population Size and Population Separation History Estimation

To estimate the effective population size and history of population separation, we applied SMC++ (v 1.15.4) [31] under 1 year per generation and a mutation rate of 1.91 × 10^−9^ per site per year [32].

## Figures and Tables

**Figure 1 ijms-26-04399-f001:**
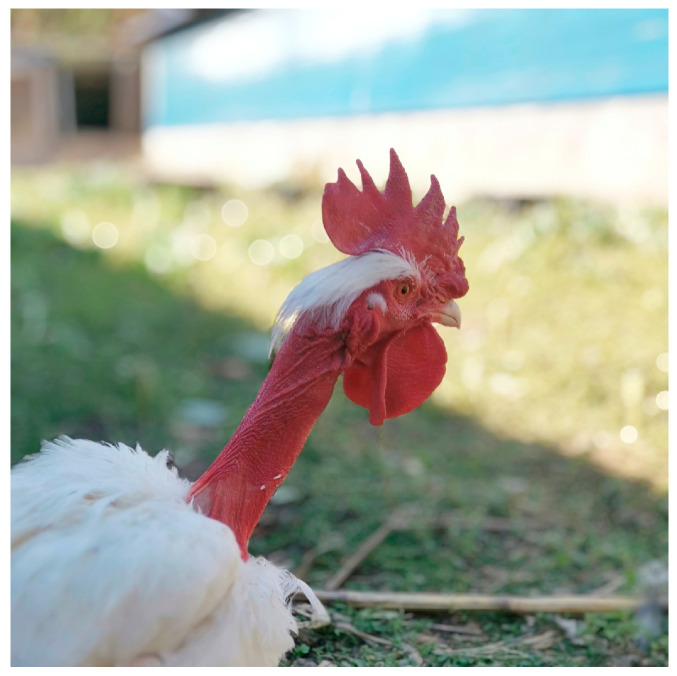
Ake chicken with the homozygous naked-neck genotype.

**Figure 2 ijms-26-04399-f002:**
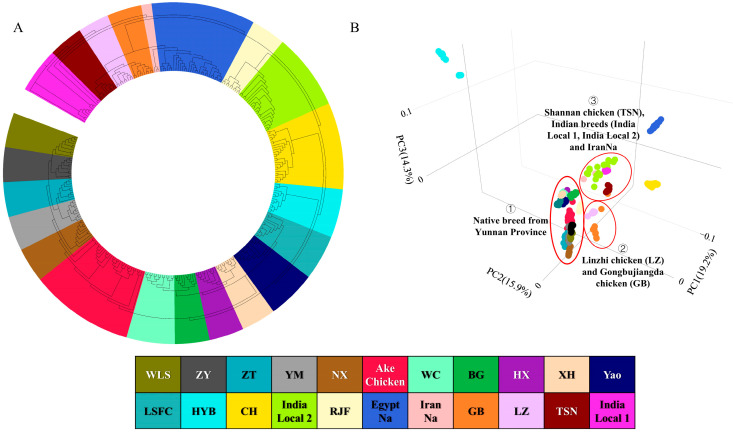
Phylogenetic relationships and population structure of 22 chicken breeds. (**A**) Neighbor-joining tree constructed from 1,022,355 autosomal SNPs. (**B**) Principal component analysis (PCA) plot based on the same SNP set, with the first three principal components explaining 19.18%, 15.86%, and 14.31% of the variance, respectively. Colors represent breed groups as in panel (**A**).2.2. Population Structure.

**Figure 3 ijms-26-04399-f003:**
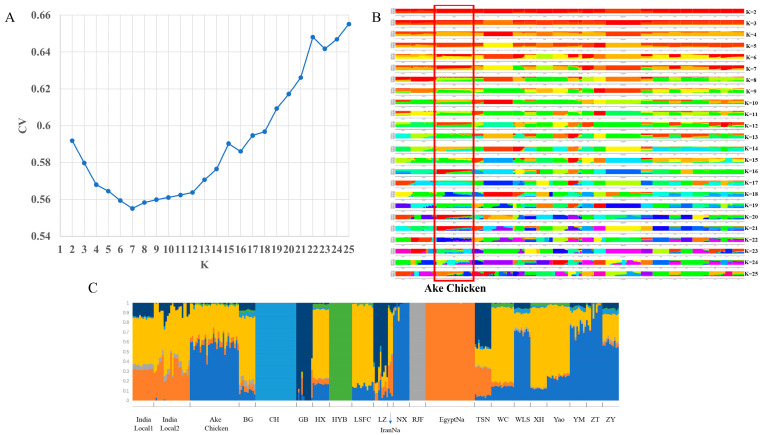
Genetic structure and ancestry analysis of chicken populations. (**A**) Cross-validation errors for K = 2 to 25 in admixture analysis. The optimal K = 7 (arrow) was selected based on the lowest error (0.42). (**B**) Ancestry proportions for K = 2 to 25. Each vertical bar represents an individual, with colors corresponding to ancestral clusters. (**C**) Ancestry composition at K = 7. Ake chickens (blue) share dominant genetic components with Yunnan breeds (e.g., NX, ZT, ZY), while showing contributions from IranNa (yellow) clusters.

**Figure 4 ijms-26-04399-f004:**
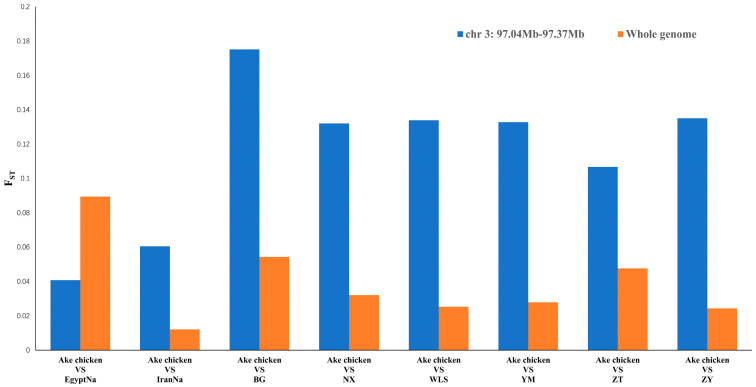
Comparison of F_ST_ values in the candidate region on chr 3: 97.0–97.37 Mb (blue bars) with genome-wide averages (orange bars).

**Figure 5 ijms-26-04399-f005:**
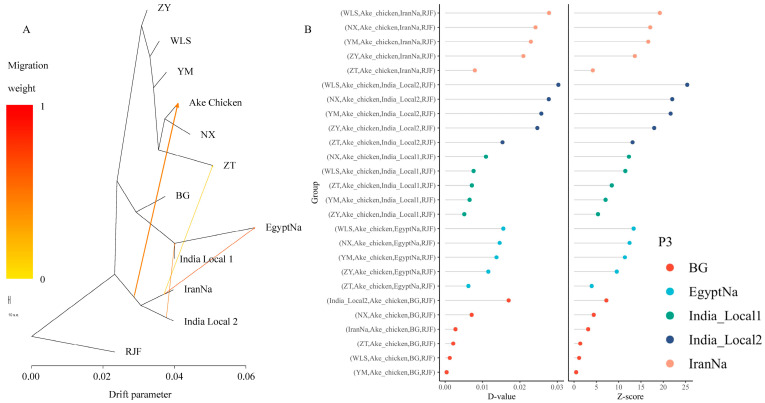
Gene flow and introgression events. (**A**) Maximum likelihood tree with 4 migration edges (arrows) inferred by TreeMix. Migration from Iranian (IranNa) and Indian (India Local 2) populations to Ake chickens is highlighted (red arrow). RJF is the outgroup. (**B**) D-statistics (|Z| > 3) showing significant gene flow between Ake chickens and Iranian/Indian breeds (positive values) but not with Egyptian naked-neck chickens (EgyptNa).

**Figure 6 ijms-26-04399-f006:**
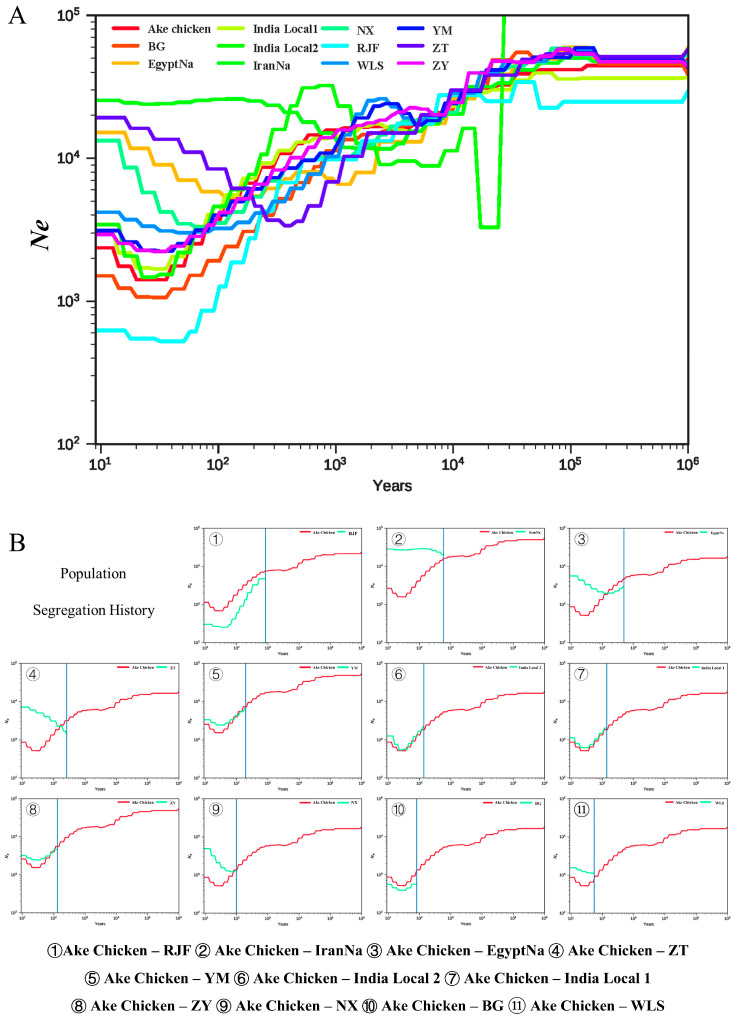
Demographic history. (**A**) Effective population size (Ne) estimated via SMC++. Ake chickens experienced a prolonged decline until recent decades (Ne ≈ 2000–3000). (**B**) Population separation history estimation. Ake chickens split from Iranian/Egyptian naked-neck breeds about 500–600 years ago and from Yunnan breeds about 100–200 years ago.

**Table 1 ijms-26-04399-t001:** Summary information of samples in this study.

Regional Classification of Breeds		Breeds	Number
Southwest China	Native breeds from Yunnan Province	Ake chicken	30
		Nixi chicken (NX)	10
		Zhaotong chicken (ZT)	10
		Yimen chicken (YM)	10
		Zhenyuan chicken (ZY)	10
		Wuliangshan chicken (WLS)	10
		Banna game chicken (BG)	10
		Chahua chicken (CH)	25
		Red jungle fowl (RJF)	10
	Native breeds from Tibet Autonomous Region	Shannan chicken (TSN)	10
		Linzhi chicken (LZ)	9
		Gongbujiangda chicken (GB)	10
	Native breeds from Guangxi Province	Yao chicken (Yao)	14
		Longsheng Feng Chicken (LSFC)	13
South China	Native breeds from Guangdong Province	Huiyang beard chicken (HYB)	14
		Huaixiang chicken (HX)	10
		Xinghua chicken (XH)	10
	Native breed from Hainan Province	Wenchang chicken (WC)	14
Other countries	Foreign breeds	Indian local chicken 1	13
		Indian local chicken 2	22
		Iranian Naked-neck chicken (IranNa)	3
		Egyptian Naked-neck chicken (EgyptNa)	30

## Data Availability

The raw sequencing data generated in this study are publicly available in the Genome Sequence Archive (GSA) of the National Genomics Data Center (NGDC) under accession number PRJCA035883 and can be found at https://ngdc.cncb.ac.cn/bioproject/browse/PRJCA035883 (accessed on 1 May 2025).

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
