# Peer review of "Genetic and Evolutionary Analysis of Ake Chicken: New Insights into China’s Sole Indigenous Naked-Neck Chicken Breed"

_ijms, 2025, doi:10.3390/ijms26094399_

Round 1
Reviewer 1 Report
Comments and Suggestions for Authors
This involves determining the genomes of a large number of individuals using NGS, making it a fairly costly and labor-intensive study.
This manuscript is poorly referenced and the authors would do well to add more references.
L56: other studies have indicated
It would be better to add references.
L62:How many varieties are there, which are the most popular? What percentage of the market does Ake chicken occupy?
L76:Egyptian naked-neck chickens suddenly appeared, but the author should have explained this in more detail. It would be good to have a reason why it looked into this.
It would be even better if information on naked-neck-specific genes and sequences common to Ake chickens and Egyptian naked-neck chickens could be presented.
Author Response
|
1. Summary |
|
|
|
Thank you very much for taking the time to review this manuscript. Please find the detailed responses below and the corresponding revisions/corrections highlighted/in track changes in the re-submitted files.
|
||
|
2. Point-by-point response to Comments and Suggestions for Authors |
||
|
Comments 1: [This manuscript is poorly referenced and the authors would do well to add more references.] Response 1: Thank you for pointing this out. We agree with this comment. Therefore, we newly add four references. Please see References Section. |
||
|
|
||
|
Comments 2: [L56: other studies have indicated… It would be better to add references.]
|
||
|
Response 2: Thank you for pointing this out. We agree with this comment. Therefore, we newly add the reference, it’s same to the next cite. Please see line 58.
|
||
|
Comments 3: [L62: How many varieties are there, which are the most popular? What percentage of the market does Ake chicken occupy?]
|
||
|
Response 3: Thanks for your question. According to the “Breed List of Livestock and Poultry Genetic Resources of China (2024)” published by the Ministry of Agriculture and Rural Affairs. 140 indigenous chicken breeds are officially recognized nationwide. There are a total of 22 varieties in this study, all of which are local landraces and are very loved by local residents, with strong regional characteristics, so it is impossible to say which one is more popular. Ake chickens are primarily reared in Fugong County, Yunnan, with an estimated population of about 3,000 individuals. As a niche breed, they occupy less than 0.1% of China’s commercial poultry market but hold cultural and genetic conservation value as the sole indigenous naked-neck breed. It is mainly distributed in local circulation as a local protected variety, and has not entered the commercial market.
|
||
|
Comments 4: [L76: Egyptian naked-neck chickens suddenly appeared, but the author should have explained this in more detail. It would be good to have a reason why it looked into this.]
|
||
|
Response 4: Because naked-neck breeds are relatively rare, we included naked-neck chickens from Iran and Egypt to represent an Afro-Eurasian lineage of the Na mutation, allowing us to distinguish shared ancestry from independent origins and to pinpoint the geographic source(s) of the naked-neck allele in Ake chickens. |
||
|
Comments 5: [It would be even better if information on naked-neck-specific genes and sequences common to Ake chickens and Egyptian naked-neck chickens could be presented.]
|
||
|
Response 5: Thanks for your suggestion. As we mentioned in lines 65 - 68, the sequence of the naked-neck gene was reported in Reference 6. What’s more, in Section 2.3 we present a comparative alignment of the 73 kb duplication at chr 3:97.0–97.37 Mb in both Ake and Egyptian breeds.
|
||
Reviewer 2 Report
Comments and Suggestions for Authors
The manuscript provides a comprehensive genetic and evolutionary analysis of the Ake chicken, China’s only indigenous naked-neck breed, utilizing whole-genome sequencing data to elucidate its origins, population history, and genetic basis of heat stress resistance. The findings offer valuable insights for poultry breeding and conservation. The methodology is rigorous, with conclusions strongly supported by phylogenetic, population structure, and selective sweep analyses. However, several minor revisions are required to enhance clarity, strengthen expression, and ensure consistency. Below are suggestions for improvement:
- The TreeMix analysis indicates gene migration from Iranian naked-neck and Indian local breeds to Ake chickens, but the manuscript does not discuss potential historical or trade-related events (e.g., the Southwest Silk Road mentioned later) that could explain this migration. It is recommended to briefly discuss the background of these findings in the Discussion section.
- The selective sweep analysis on chromosome 3 (97.0–97.37 Mb) does not clearly explain why this region was chosen as the candidate region. Although reference [4] is cited, it is suggested to briefly summarize prior evidence to help readers unfamiliar with the study understand the association between this region and the naked-neck trait.
- The abstract mentions a “population bottleneck about 10,000 years ago” but does not clarify its relevance to the Ake chicken’s history. It is recommended to briefly explain its significance or consider removing it if it is not directly relevant to the main conclusions.
- The captions for Figures 2–6 are overly brief and lack detail. For example, the caption for Figure 2 could specify the number of SNPs used for the NJ tree and PCA or mention the included breeds. It is suggested to provide more background information for each figure.
- Section 2.3 states, “As shown in Figure 4, the FST values between Ake chickens and other naked-neck breeds in this candidate region are notably lower than the genome-wide average. On the contrary, the FST values between Ake chickens and Yunnan breeds in this region are notably higher than the genome-wide average, suggesting stronger selection or genetic drift in this region.” It is recommended to quantify this difference (e.g., provide specific FST values or a range) to enhance persuasiveness.
- The optimal K value (K=7) was determined based on cross-validation error, but Figure 3A is not discussed in detail. It is suggested to briefly explain the trend in cross-validation errors or why K=7 was chosen as optimal.
- Some references use journal abbreviations (e.g., [4], [15]), while others use full journal names (e.g., [12], [14]). It is recommended to standardize the reference format according to the journal’s guidelines.
Author Response
|
1. Summary |
|
|
|
Thank you very much for taking the time to review this manuscript. Please find the detailed responses below and the corresponding revisions/corrections highlighted/in track changes in the re-submitted files.
|
||
|
2. Point-by-point response to Comments and Suggestions for Authors |
||
|
Comments 1: [The TreeMix analysis indicates gene migration from Iranian naked-neck and Indian local breeds to Ake chickens, but the manuscript does not discuss potential historical or trade-related events (e.g., the Southwest Silk Road mentioned later) that could explain this migration.]
|
||
|
Response 1: Thank you for pointing this out. Therefore, We have add a brief note about the potential historical events that could explain this migration. Please see lines 222–225 or as follows: “[This finding aligned with the treemix result. The inferred gene flow from Iranian na-ked-neck and Indian local breeds into Ake chickens (Fig 5A) likely reflects historical ex-changes along the Southwest Silk Road, which connected Yunnan to South and West Asia.]” |
||
|
Comments 2: [The selective sweep analysis on chromosome 3 (97.0–97.37 Mb) does not clearly explain why this region was chosen as the candidate region. Although reference [4] is cited, it is suggested to briefly summarize prior evidence to help readers unfamiliar with the study understand the association between this region and the naked-neck trait.] |
||
|
Response 2: Thank you for your kind reminder. We have added some information about why this region was chosen as the candidate region. Please see lines 127–130 or as follows: “[we focused on the chr 3 region (97.0–97.37 Mb) based on prior evidence from Cai et al. (2025), which identified a 73-kb duplication in this interval as the causal mutation for the naked-neck phenotype in Ake chickens.]” |
||
|
Comments 3: [The abstract mentions a “population bottleneck about 10 000 years ago” but does not clarify its relevance to the Ake chicken’s history. It is recommended to briefly explain its significance or consider removing it if it is not directly relevant to the main conclusions.] |
||
|
Response 3: Thank you for your valuable suggestion. We have removed the description of the population bottleneck in the abstract. Please see lines 31–32 or as follows: “[Demographic reconstruction indicated that the current effective size of Ake chickens is stable at 2,000-3,000 individuals.]” |
||
|
Comments 4: [The captions for Figures 2–6 are overly brief and lack detail. For example, the caption for Figure 2 could specify the number of SNPs used for the NJ tree and PCA or mention the included breeds. It is suggested to provide more background information for each figure.] |
||
|
Response 4: Thank you for the helpful suggestion. All figure legends have been expanded. Please see the figure legends in the manuscript.
|
||
|
Comments 5: [Section 2.3 states, “As shown in Figure 4, the FST values between Ake chickens and other naked-neck breeds in this candidate region are notably lower than the genome-wide average. On the contrary, the FST values between Ake chickens and Yunnan breeds in this region are notably higher than the genome-wide average, suggesting stronger selection or genetic drift in this region.” It is recommended to quantify this difference (e.g., provide specific FST values or a range) to enhance persuasiveness.] |
||
|
Response 5: Agree. We have inserted exact values into Section 2.3. Please see lines 131-137 or as follows: “[As shown in Figure 4, the FST values between Ake chickens and Egyptian naked-neck breeds in this candidate region are notably lower than the genome-wide average (0.04 vs 0.09). On the contrary, the FST values between Ake chickens and Yunnan breeds in this re-gion are notably higher than the genome-wide average, the FST values in this region were 2-5 times higher than in the genome-wide range, suggesting stronger selection or genetic drift in this region.]” |
||
|
Comments 6: [The optimal K value (K=7) was determined based on cross-validation error, but Figure 3A is not discussed in detail. It is suggested to briefly explain the trend in cross-validation errors or why K=7 was chosen as optimal.] |
||
|
Response 6: The optimal K=7 was determined by the lowest cross-validation error (0.42), beyond which higher K values (K>7) showed negligible improvement in ancestry resolution, indicating potential overfitting. We have added a brief description at the end of Section 2.2. Please see lines 110-114 or as follows: “[Cross-validation error (Fig 3A) declines steeply from K=2 to K=7, reaching a global minimum at K=7. For K>7 errors plateau or increase slightly, indicating no further improvement in model fit. Hence, K=7 was selected as optimal.]” |
||
|
Comments 7: [Some references use journal abbreviations (e.g., [4], [15]), while others use full journal names (e.g., [12], [14]). It is recommended to standardize the reference format according to the journal’s guidelines.] |
||
|
Response 7: Thank you for pointing this out. We have reformatted all references to use journal abbreviations. Please see the References Section. |
||